# MEASURING MASSIVE MULTITASK LANGUAGE UNDERSTANDING

**Dan Hendrycks**
UC Berkeley

**Collin Burns**
Columbia University

**Steven Basart**
UChicago

**Andy Zou**
UC Berkeley

**Mantas Mazeika**
UIUC

**Dawn Song**
UC Berkeley

**Jacob Steinhardt**
UC Berkeley

## ABSTRACT

We propose a new test to measure a text model's multitask accuracy. The test covers 57 tasks including elementary mathematics, US history, computer science, law, and more. To attain high accuracy on this test, models must possess extensive world knowledge and problem solving ability. We find that while most recent models have near random-chance accuracy, the very largest GPT-3 model improves over random chance by almost 20 percentage points on average. However, on every one of the 57 tasks, the best models still need substantial improvements before they can reach expert-level accuracy. Models also have lopsided performance and frequently do not know when they are wrong. Worse, they still have near-random accuracy on some socially important subjects such as morality and law. By comprehensively evaluating the breadth and depth of a model's academic and professional understanding, our test can be used to analyze models across many tasks and to identify important shortcomings.

## 1 INTRODUCTION

Natural Language Processing (NLP) models have achieved superhuman performance on a number of recently proposed benchmarks. However, these models are still well below human level performance for language understanding as a whole, suggesting a disconnect between our benchmarks and the actual capabilities of these models. The General Language Understanding Evaluation benchmark (GLUE) (Wang et al., 2018) was introduced in 2018 to evaluate performance on a wide range of NLP tasks, and top models achieved superhuman performance within a year. To address the shortcomings of GLUE, researchers designed the SuperGLUE benchmark with more difficult tasks (Wang et al., 2019). About a year since the release of SuperGLUE, performance is again essentially human-level (Raffel et al., 2019). While these benchmarks evaluate linguistic skills more than overall language understanding, an array of commonsense benchmarks have been proposed to measure basic reasoning and everyday knowledge (Zellers et al., 2019; Huang et al., 2019; Bisk et al., 2019). However, these recent benchmarks have similarly seen rapid progress (Khashabi et al., 2020). Overall, the near human-level performance on these benchmarks suggests that they are not capturing important facets of language understanding.

Transformer models have driven this recent progress by pretraining on massive text corpora, including all of Wikipedia, thousands of books, and numerous websites. These models consequently see extensive information about specialized topics, most of which is not assessed by existing NLP benchmarks. It consequently remains an open question just how capable current language models are at learning and applying knowledge from many domains.

To bridge the gap between the wide-ranging knowledge that models see during pretraining and the existing measures of success, we introduce a new benchmark for assessing models across a diverse set of subjects that humans learn. We design the benchmark to measure knowledge acquired during pretraining by evaluating models exclusively in zero-shot and few-shot settings. This makes the benchmark more challenging and more similar to how we evaluate humans. The benchmark covers 57 subjects across STEM, the humanities, the social sciences, and more. It ranges in difficulty from an elementary level to an advanced professional level, and it tests both world knowledge and problem solving ability. Subjects range from traditional areas, such as mathematics and history, to more

| Few Shot Prompt and Predicted Answer |
| --- |

The following are multiple choice questions about high school mathematics.

How many numbers are in the list 25, 26, ..., 100? (A) 75 (B) 76 (C) 22 (D) 23
Answer: B

Compute $i + i^2 + i^3 + \cdots + i^{258} + i^{259}$.
(A) -1 (B) 1 (C) $i$ (D) -$i$
Answer: A

If 4 daps = 7 yaps, and 5 yaps = 3 baps, how many daps equal 42 baps?
(A) 28 (B) 21 (C) 40 (D) 30
Answer: C

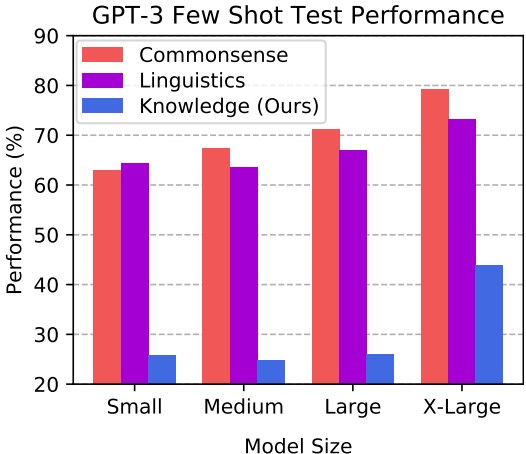

(a) An example of few-shot learning and inference using GPT-3. The blue underlined bold text is the auto-completed response from GPT-3, while the preceding text is the user-inputted prompt. In this 2-shot learning example, there are two instruction examples and one initially incomplete example. On average, GPT-3 has low accuracy on high school mathematics questions.

(b) Performance on a commonsense benchmark (HellaSwag), a linguistic understanding benchmark (SuperGLUE), and the massive multitask test. On previous benchmarks, smaller models start well above random chance levels and exhibit more continuous improvements with model size increases, but on our test, GPT-3 moves beyond random chance with the largest model.

specialized areas like law and ethics (Hendrycks et al., 2020). The granularity and breadth of the subjects makes the benchmark ideal for identifying a model's blind spots.

We find that meaningful progress on our benchmark has only become possible in recent months. In particular, few-shot models up to 13 billion parameters (Brown et al., 2020) achieve random chance performance of 25% accuracy, but the 175 billion parameter GPT-3 model reaches a much higher 43.9% accuracy (see Figure 1b). On the other hand, unlike human professionals GPT-3 does not excel at any single subject. Instead, we find that performance is lopsided, with GPT-3 having almost 70% accuracy for its best subject but near-random performance for several other subjects.

Our results indicate that while recent advances have been impressive, state-of-the-art models still struggle at learning and applying knowledge from pretraining. The tasks with near-random accuracy include calculation-heavy subjects such as physics and mathematics and subjects related to human values such as law and morality. This second weakness is particularly concerning because it will be important for future models to have a strong understanding of what is legal and what is ethical. Worryingly, we also find that GPT-3 does not have an accurate sense of what it does or does not know since its average confidence can be up to 24% off from its actual accuracy. We comprehensively evaluate the breadth and depth of a model's text understanding by covering numerous topics that humans are incentivized to learn. Since our test consists in 57 tasks, it can be used to analyze aggregate properties of models across tasks and to track important shortcomings. The test and code is available at github.com/hendrycks/test.

## 2 RELATED WORK

**Pretraining.** The dominant paradigm in NLP is to pretrain large models on massive text corpora including educational books and websites. In the process, these models are exposed to information about a wide range of topics. Petroni et al. (2019) found that recent models learn enough information from pretraining that they can serve as knowledge bases. However, no prior work has comprehensively measured the knowledge models have across many real-world domains.

Until recently, researchers primarily used fine-tuned models on downstream tasks (Devlin et al., 2019). However, larger pretrained models like GPT-3 (Brown et al., 2020) have made it possible to achieve competitive performance without fine-tuning by using few-shot learning, which removes the need for a large fine-tuning set. With the advent of strong zero-shot and few-shot learning, it is now possible to curate a diverse set of tasks for evaluation and remove the possibility of models on "spurious cues" (Geirhos et al., 2020; Hendrycks et al., 2019b) in a dataset to achieve high performance.

**Benchmarks.** Many recent benchmarks aim to assess a model's general world knowledge and basic reasoning ability by testing its "commonsense." A number of commonsense benchmarks have been

**Professional Law**

As Seller, an encyclopedia salesman, approached the grounds on which Hermit's house was situated, he saw a sign that said, "No salesmen. Trespassers will be prosecuted. Proceed at your own risk." Although Seller had not been invited to enter, he ignored the sign and drove up the driveway toward the house. As he rounded a curve, a powerful explosive charge buried in the driveway exploded, and Seller was injured. Can Seller recover damages from Hermit for his injuries?
(A) Yes, unless Hermit, when he planted the charge, intended only to deter, not harm, intruders. ✘
(B) Yes, if Hermit was responsible for the explosive charge under the driveway. ✔
(C) No, because Seller ignored the sign, which warned him against proceeding further. ✘
(D) No, if Hermit reasonably feared that intruders would come and harm him or his family. ✘

Figure 2: This task requires understanding detailed and dissonant scenarios, applying appropriate legal precedents, and choosing the correct explanation. The green checkmark is the ground truth.

proposed in the past year, but recent models are already nearing human-level performance on several of these, including HellaSwag (Zellers et al., 2019), Physical IQA (Bisk et al., 2019), and CosmosQA (Huang et al., 2019). By design, these datasets assess abilities that almost every child has. In contrast, we include harder specialized subjects that people must study to learn.

Some researchers have suggested that the future of NLP evaluation should focus on Natural Language Generation (NLG) (Zellers et al., 2020), an idea that reaches back to the Turing Test (Turing, 1950). However, NLG is notoriously difficult to evaluate and lacks a standard metric (Sai et al., 2020). Consequently, we instead create a simple-to-evaluate test that measures classification accuracy on multiple choice questions.

While several question answering benchmarks exist, they are comparatively limited in scope. Most either cover easy topics like grade school subjects for which models can already achieve strong performance (Clark et al., 2018; Khot et al., 2019; Mihaylov et al., 2018; Clark et al., 2019), or are focused on linguistic understanding in the form of reading comprehension (Lai et al., 2017; Richardson et al., 2013). In contrast, we include a wide range of difficult subjects that go far beyond linguistic understanding.

## 3 A MULTITASK TEST

We create a massive multitask test consisting of multiple-choice questions from various branches of knowledge. The test spans subjects in the humanities, social sciences, hard sciences, and other areas that are important for some people to learn. There are 57 tasks in total, which is also the number of Atari games (Bellemare et al., 2013), all of which are listed in Appendix B. The questions in the dataset were manually collected by graduate and undergraduate students from freely available sources online. These include practice questions for tests such as the Graduate Record Examination and the United States Medical Licensing Examination. It also includes questions designed for undergraduate courses and questions designed for readers of Oxford University Press books. Some tasks cover a subject, like psychology, but at a specific level of difficulty, such as "Elementary," "High School," "College," or "Professional." For example, the "Professional Psychology" task draws on questions from freely available practice questions for the Examination for Professional Practice in Psychology, while the "High School Psychology" task has questions like those from Advanced Placement Psychology examinations.

We collected 15908 questions in total, which we split into a few-shot development set, a validation set, and a test set. The few-shot development set has 5 questions per subject, the validation set may be used for selecting hyperparameters and is made of 1540 questions, and the test set has 14079 questions. Each subject contains 100 test examples at the minimum, which is longer than most exams designed to assess people.

Human-level accuracy on this test varies. Unspecialized humans from Amazon Mechanical Turk obtain 34.5% accuracy on this test. Meanwhile, expert-level performance can be far higher. For example, real-world test-taker human accuracy at the 95th percentile is around 87% for US Medical Licensing Examinations, and these questions make up our "Professional Medicine" task. If we take the 95th percentile human test-taker accuracy for exams that build up our test, and if we make an educated guess when such information is unavailable, we then estimate that expert-level accuracy is approximately 89.8%.

Since our test aggregates different subjects and several levels of difficulty, we measure more than straightforward commonsense or narrow *linguistic* understanding. Instead, we measure arbitrary

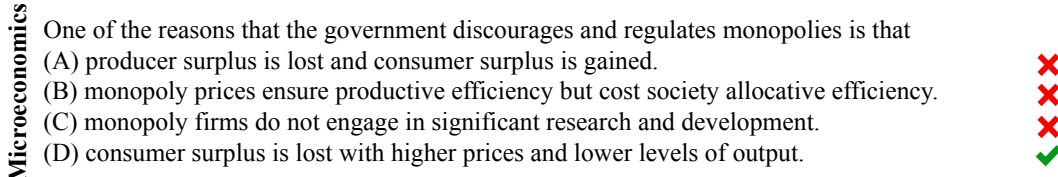

Figure 3: Examples from the Microeconomics task.

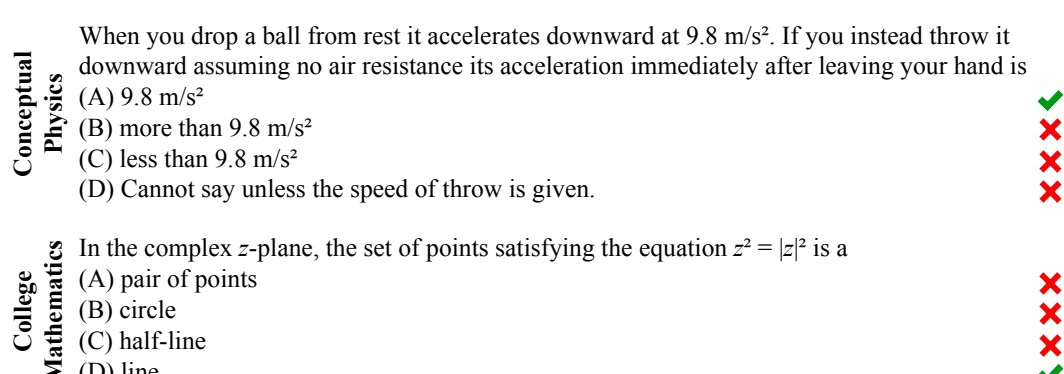

Figure 4: Examples from the Conceptual Physics and College Mathematics STEM tasks.

real-world *text* understanding. Since models are pretrained on the Internet, this enables us to test how well they can extract useful knowledge from massive corpora. Future models that use this test could be single models or a mixture of experts model. To succeed at our test, future models should be well-rounded, possess extensive world knowledge, and develop expert-level problem solving ability. These properties make the test likely to be an enduring and informative goalpost.

### 3.1 HUMANITIES

The humanities is a group of disciplines that make use of qualitative analysis and analytic methods rather than scientific empirical methods. Branches of the humanities include law, philosophy, history, and so on (Appendix B). Mastering these subjects requires a variety of skills. For example, legal understanding requires knowledge of how to apply rules and standards to complex scenarios, and also provide answers with stipulations and explanations. We illustrate this in Figure 2. Legal understanding is also necessary for understanding and following rules and regulations, a necessary capability to constrain open-world machine learning models. For philosophy, our questions cover concepts like logical fallacies, formal logic, and famous philosophical arguments. It also covers moral scenarios, including questions from the ETHICS dataset (Hendrycks et al., 2020) that test a model's understanding of normative statements through predicting widespread moral intuitions about diverse everyday scenarios. Finally, our history questions cover a wide range of time periods and geographical locations, including prehistory and other advanced subjects.

### 3.2 SOCIAL SCIENCE

Social science includes branches of knowledge that examine human behavior and society. Subject areas include economics, sociology, politics, geography, psychology, and so on. See Figure 3 for an example question. Our economics questions include microeconomics, macroeconomics, and econometrics, and cover different types of problems, including questions that require a mixture of world knowledge, qualitative reasoning, or quantitative reasoning. We also include important but more esoteric topics such as security studies in order to test the boundaries of what is experienced and learned during pretraining. Social science also includes psychology, a field that may be especially important for attaining a nuanced understanding of humans.

### 3.3 SCIENCE, TECHNOLOGY, ENGINEERING, AND MATHEMATICS (STEM)

STEM subjects include physics, computer science, mathematics, and more. Two examples are shown in Figure 4. Conceptual physics tests understanding of simple physics principles and may be thought

**Professional Medicine**

A 33-year-old man undergoes a radical thyroidectomy for thyroid cancer. During the operation, moderate hemorrhaging requires ligation of several vessels in the left side of the neck. Postoperatively, serum studies show a calcium concentration of 7.5 mg/dL, albumin concentration of 4 g/dL, and parathyroid hormone concentration of 200 pg/mL. Damage to which of the following vessels caused the findings in this patient?
(A) Branch of the costocervical trunk      ✖
(B) Branch of the external carotid artery      ✖
(C) Branch of the thyrocervical trunk      ✔
(D) Tributary of the internal jugular vein      ✖

Figure 5: A question from the Professional Medicine task.

of as a harder version of the physical commonsense benchmark Physical IQA (Bisk et al., 2019). We also test mathematical problem solving ability at various levels of difficulty, from the elementary to the college level. College mathematics questions, like those found on the GRE mathematics subject test, often require chains of reasoning and abstract knowledge. To encode mathematics expressions, we use LaTeX or symbols such as * and ^ for multiplication and exponentiation respectively. STEM subjects require knowledge of empirical methods, fluid intelligence, and procedural knowledge.

## 3.4 OTHER

There is a long tail of subjects that either do not neatly fit into any of the three preceding categories or for which there are not thousands of freely available questions. We put these subjects into Other. This section includes the Professional Medicine task, which has difficult questions that require humans many years of study to master. An example is depicted in Figure 5. This section also contains business topics like finance, accounting, and marketing, as well as knowledge of global facts. The latter includes statistics about poverty in different countries over time, which may be necessary for having an accurate model of the world internationally.

## 4 EXPERIMENTS

### 4.1 SETUP

**Assessment and Models.**   To measure performance on our multitask test, we compute the classification accuracy across all examples and tasks. We evaluate GPT-3 (Brown et al., 2020) and UnifiedQA (Khashabi et al., 2020). For GPT-3 we use the OpenAI API, which provides access to four model variants, "Ada," "Babbage," "Curie," and "Davinci," which we refer to as "Small" (2.7 billion parameters), "Medium" (6.7 billion), "Large" (13 billion) and "X-Large" (175 billion). UnifiedQA uses the T5 (Raffel et al., 2019) text-to-text backbone and is fine-tuned on previously proposed question answering datasets (Lai et al., 2017), where the prediction is the class with the highest token overlap with UnifiedQA's text output. Since UnifiedQA is fine-tuned on other datasets, we evaluate it without any further tuning to assess its transfer accuracy. We also fine-tune RoBERTa-base, ALBERT-xxlarge, and GPT-2 on UnifiedQA training data and our dev+val set. We primarily focus on UnifiedQA and GPT-3 in the rest of this document, but additional discussion of RoBERTa, ALBERT, and GPT-2 is in Appendix A.

| Model | Humanities | Social Science | STEM | Other | Average |
|---|---|---|---|---|---|
| Random Baseline | 25.0 | 25.0 | 25.0 | 25.0 | 25.0 |
| RoBERTa | 27.9 | 28.8 | 27.0 | 27.7 | 27.9 |
| ALBERT | 27.2 | 25.7 | 27.7 | 27.9 | 27.1 |
| GPT-2 | 32.8 | 33.3 | 30.2 | 33.1 | 32.4 |
| UnifiedQA | 45.6 | 56.6 | 40.2 | 54.6 | 48.9 |
| GPT-3 Small (few-shot) | 24.4 | 30.9 | 26.0 | 24.1 | 25.9 |
| GPT-3 Medium (few-shot) | 26.1 | 21.6 | 25.6 | 25.5 | 24.9 |
| GPT-3 Large (few-shot) | 27.1 | 25.6 | 24.3 | 26.5 | 26.0 |
| GPT-3 X-Large (few-shot) | 40.8 | 50.4 | 36.7 | 48.8 | 43.9 |

Table 1: Average weighted accuracy for each model on all four broad disciplines. All values are percentages. Some models proposed in the past few months can move several percent points beyond random chance. GPT-3 uses few-shot learning and UnifiedQA is tested under distribution shift.

**Few-Shot Prompt.** We feed GPT-3 prompts like that shown in Figure 1a. We begin each prompt with "The following are multiple choice questions (with answers) about [subject]." For zero-shot evaluation, we append the question to the prompt. For few-shot evaluation, we add up to 5 demonstration examples with answers to the prompt before appending the question. All prompts end with "Answer: ". The model then produces probabilities for the tokens "A," "B," "C," and "D," and we treat the highest probability option as the prediction. For consistent evaluation, we create a dev set with 5 fixed few-shot examples for each subject.

## 4.2 RESULTS

**Model Size and Accuracy.** We compare the few-shot accuracy of each GPT-3 size in Table 1. We find that the three smaller GPT-3 models have near random accuracy (around 25%). In contrast, we find that the X-Large 175 billion parameter GPT-3 model performs substantially better than random, with an accuracy of 43.9%. We also find qualitatively similar results in the zero-shot setting. While the smaller models have around 25% zero-shot accuracy, Figure 10 in Appendix A shows that the largest GPT-3 model has a much higher zero-shot accuracy of about 37.7%. Brown et al. (2020) also observe that larger GPT-3 models perform better, though progress tends to be steadier. In Figure 1b we show that non-random accuracy on the multitask test emerged with recent large few-shot models compared to datasets that assess commonsense and linguistic understanding.

To test the usefulness of fine-tuning instead of few-shot learning, we also evaluate UnifiedQA models. UnifiedQA has the advantage of being fine-tuned on other question answering datasets, unlike GPT-3. We assess UnifiedQA by evaluating its transfer performance without any additional fine-tuning. The largest UnifiedQA model we test has 11 billion parameters, which is slightly smaller than GPT-3 Large. Neverthe-less, we show in Table 1 that it attains 48.9%

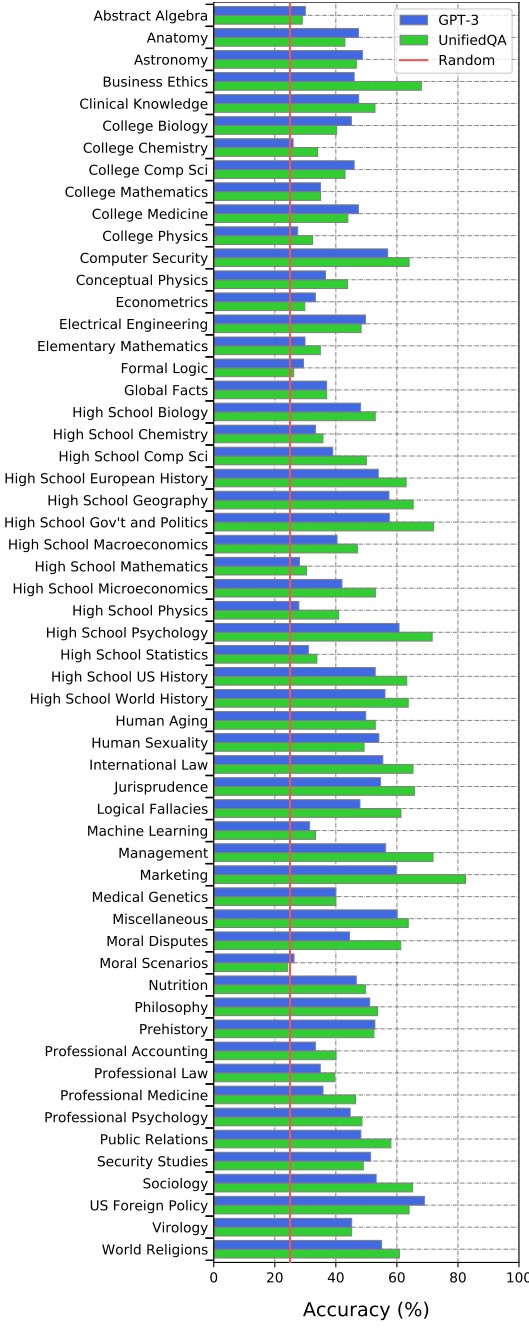

Figure 6: GPT-3 (few-shot) and UnifiedQA results.

accuracy. This performs better than the few-shot GPT-3 X-Large model, despite UnifiedQA have an order of magnitude fewer parameters. We also find that even the smallest UnifiedQA variant, with just 60 million parameters, has approximately 29.3% accuracy. These results suggest that while model size is a key component for achieving strong performance, fine-tuning also helps.

**Comparing Disciplines.** Using our test, we discover that GPT-3 and UnifiedQA have lopsided performance and several substantial knowledge gaps. Figure 6 shows the accuracy of GPT-3 (few-shot) and UnifiedQA for all 57 tasks. It shows the both models are below expert-level performance for all tasks, with GPT-3's accuracy ranging from 69% for US Foreign Policy to 26% for College Chemistry. UnifiedQA does best on marketing, with an accuracy of 82.5%.

Overall, models do poorly on highly procedural problems. Figure 6 shows that calculation-heavy STEM subjects tend to have low accuracy compared to verbal subjects. For GPT-3, 9 out of the 10

Declarative vs. Procedural Knowledge

Prompt and Completion:

The order of operations or PEMDAS is
**Parentheses Exponents Multiplication Division Addition Subtraction**

- - - - - - - - - - - - - - - - - - - - - - -

Prompt and Completion:

$(1 + 1) \times 2 = $ **3**

Figure 7: GPT-3's completion for two prompts testing knowledge of the order of operations. The blue underlined bold text is the autocompleted response from GPT-3. While it *knows about* the order of operations, it sometimes does not *know how* to apply its knowledge.

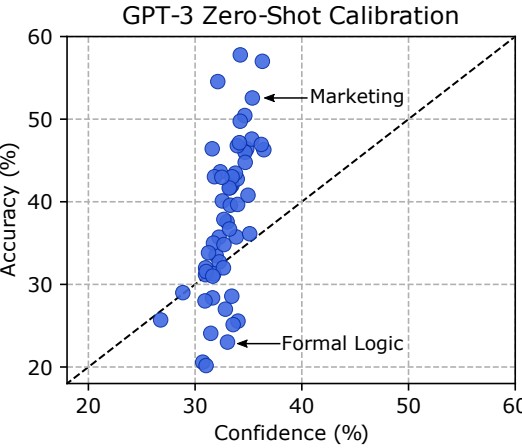

Figure 8: GPT-3's confidence is a poor estimator of its accuracy and can be off by up to 24%.

lowest-accuracy tasks are STEM subjects that emphasize mathematics or calculations. We speculate that is in part because GPT-3 acquires declarative knowledge more readily than procedural knowledge. For example, many questions in Elementary Mathematics require applying the order of operations for arithmetic, which is described by the acronym PEMDAS (Parentheses Exponents Multiplication Division Addition Subtraction). In Figure 7, we confirm that GPT-3 is *aware* of the acronym PEMDAS. However, it does not consistently *apply* PEMDAS to actual problems. On the other hand, procedural understanding is not its only weak point. We find that some verbal tasks such as Moral Scenarios from Hendrycks et al. (2020) and Professional Law also have especially low accuracy.

Our test also shows that GPT-3 acquires knowledge quite unlike humans. For example, GPT-3 learns about topics in a pedagogically unusual order. GPT-3 does better on College Medicine (47.4%) and College Mathematics (35.0%) than calculation-heavy Elementary Mathematics (29.9%). GPT-3 demonstrates unusual breadth, but it does not master a single subject. Meanwhile we suspect humans have mastery in several subjects but not as much breadth. In this way, our test shows that GPT-3 has many knowledge blindspots and has capabilities that are lopsided.

**Calibration.** We should not trust a model's prediction unless the model is calibrated, meaning that its confidence is a good estimate of the actual probability the prediction is correct. However, large neural networks are often miscalibrated (Guo et al., 2017), especially under distribution shift (Ovadia et al., 2019). We evaluate the calibration of GPT-3 by testing how well its average confidence estimates its actual accuracy for each subject. We show the results in Figure 8, which demonstrates that GPT-3 is uncalibrated. In fact, its confidence is only weakly related to its actual accuracy in the zero-shot setting, with the difference between its accuracy and confidence reaching up to 24% for some subjects. Another calibration measure is the Root Mean Squared (RMS) calibration error (Hendrycks et al., 2019a; Kumar et al., 2019). Many tasks have miscalibrated predictions, such as Elementary Mathematics which has a zero-shot RMS calibration error of 19.4%. Models are only somewhat more calibrated in the few-shot setting, as shown in Appendix A. These results suggest that model calibration has wide room for improvement.

## 5    DISCUSSION

**Multimodal Understanding.** While text is capable of conveying an enormous number of concepts about the world, many important concepts are conveyed mainly through other modalities, such as images, audio, and physical interaction (Bisk et al., 2020). Existing large-scale NLP models, such as GPT-3, do not incorporate multimodal information, so we design our benchmark to capture a diverse array of tasks in a text-only format. However, as models gain the ability to process multimodal inputs, benchmarks should be designed to reflect this change. One such benchmark could be a "Turk Test," consisting of Amazon Mechanical Turk Human Intelligence Tasks. These are well-defined tasks that require models to interact with flexible formats and demonstrate multimodal understanding.

**The Internet as a Training Set.** A major distinction between our benchmark and previous multitask NLP benchmarks is that we do not require large training sets. Instead, we assume that models have acquired the requisite knowledge from reading vast quantities of diverse text from the Internet. This

process is typically called pretraining, but it can be thought of as training in its own right, where the downstream evaluation is demonstrating whatever knowledge we would expect a human to pick up from reading the same text.

This motivates us to propose a methodological change so that models are trained more like how humans learn. While most previous machine learning benchmarks have models learn from a large question bank, humans primarily learn new subjects by reading books and listening to others talk about the topic. For specialized subjects such as Professional Law, massive legal corpora are available, such as the 164-volume legal encyclopedia *Corpus Juris Secundum*, but there are fewer than 5,000 multistate bar exam questions available. Learning the entire law exclusively through a small number of practice tests is implausible, so future models must learn more during pretraining.

For this reason we assess pretrained models in a zero-shot, few-shot, or transfer setting and we provide a dev, val, and test set for each task. The dev set is used for few-shot prompts, the val set could be used for hyperparameter tuning, and the test set is used to compute the final accuracy. Importantly, the format of our evaluation is not identical to the format in which information is acquired during pretraining. This has the benefit of obviating concerns about spurious training set annotation artifacts (Geirhos et al., 2020; Hendrycks et al., 2019b) and is in stark contrast to the previous paradigm of identically distributed training and test sets. This change also enables collecting a much more extensive and diverse set of tasks for evaluation. We anticipate our methodology becoming more widespread as models improve at extracting information from diverse online sources.

**Model Limitations.** We find that current large-scale Transformers have wide room for improvement. They are notably poor at modeling human (dis)approval, as evident by the low performance on the Professional Law and Moral Scenarios tasks. For future systems to be aligned with human values, high performance on these tasks is crucial (Hendrycks et al., 2020), so future research should especially aim to increase accuracy on these tasks. Models also have difficulty performing calculations, so much so that they exhibit poor performance on Elementary Mathematics and many other STEM subjects with "plug and chug" problems. Additionally, they do not match expert-level performance (90%) on any subject, so for all subjects it is subhuman. On average, models are only now starting to move beyond random-chance accuracy levels.

Addressing these shortcomings may be challenging. To illustrate this, we attempted to create a better Professional Law model by pretraining on specialized data but achieved only limited success. We collected approximately 2,000 additional Professional Law training examples. After fine-tuning a RoBERTa-base model (Liu et al., 2019) using this custom training set, our model attained 32.8% test accuracy. To test the impact of additional specialized training data, we also had RoBERTa continue pretraining on approximately 1.6 million legal case summaries using Harvard's Law Library case law corpus `case.law`, but after fine-tuning it only attained 36.1% accuracy. This suggests that while additional pretraining on relevant high quality text can help, it may not be enough to substantially increase the performance of current models.

It is unclear whether simply scaling up existing language models will solve the test. Current understanding indicates that a $10\times$ increase in model size must be accompanied by an approximate $5\times$ increase in data (Kaplan et al., 2020). Aside from the tremendous expense in creating multi-trillion parameter language models, data may also become a bottleneck, as there is far less written about esoteric branches of knowledge than about everyday situations.

## 6 CONCLUSION

We introduced a new test that measures how well text models can learn and apply knowledge encountered during pretraining. By covering 57 subjects at varying levels of difficulty, the test assesses language understanding in greater breadth and depth than previous benchmarks. We found that it has recently become possible for models to make meaningful progress on the test, but that state-of-the-art models have lopsided performance and rarely excel at any individual task. We also showed that current models are uncalibrated and have difficulty with tasks that require calculations. Worryingly, models also perform especially poorly on socially relevant subjects including morality and law. Our expansive test can help researchers pinpoint important shortcomings of models, making it easier to gain a clearer picture of state-of-the-art capabilities.

## ACKNOWLEDGEMENTS

We would like to thank the following for their helpful comments: Oyvind Tafjord, Jan Leike, David Krueger, Alex Tamkin, Girish Sastry, and Henry Zhu. DH is supported by the NSF GRFP Fellowship and an Open Philanthropy Project Fellowship. This research was also supported by the NSF Frontier Award 1804794.

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

# A    ADDITIONAL ANALYSIS

This appendix includes figures with sorted results (Figure 9), few-shot examples vs. accuracy (Figure 10), and few-shot calibration (Figure 11). It also includes sections on fine-tuning, error analysis, and format sensitivity.

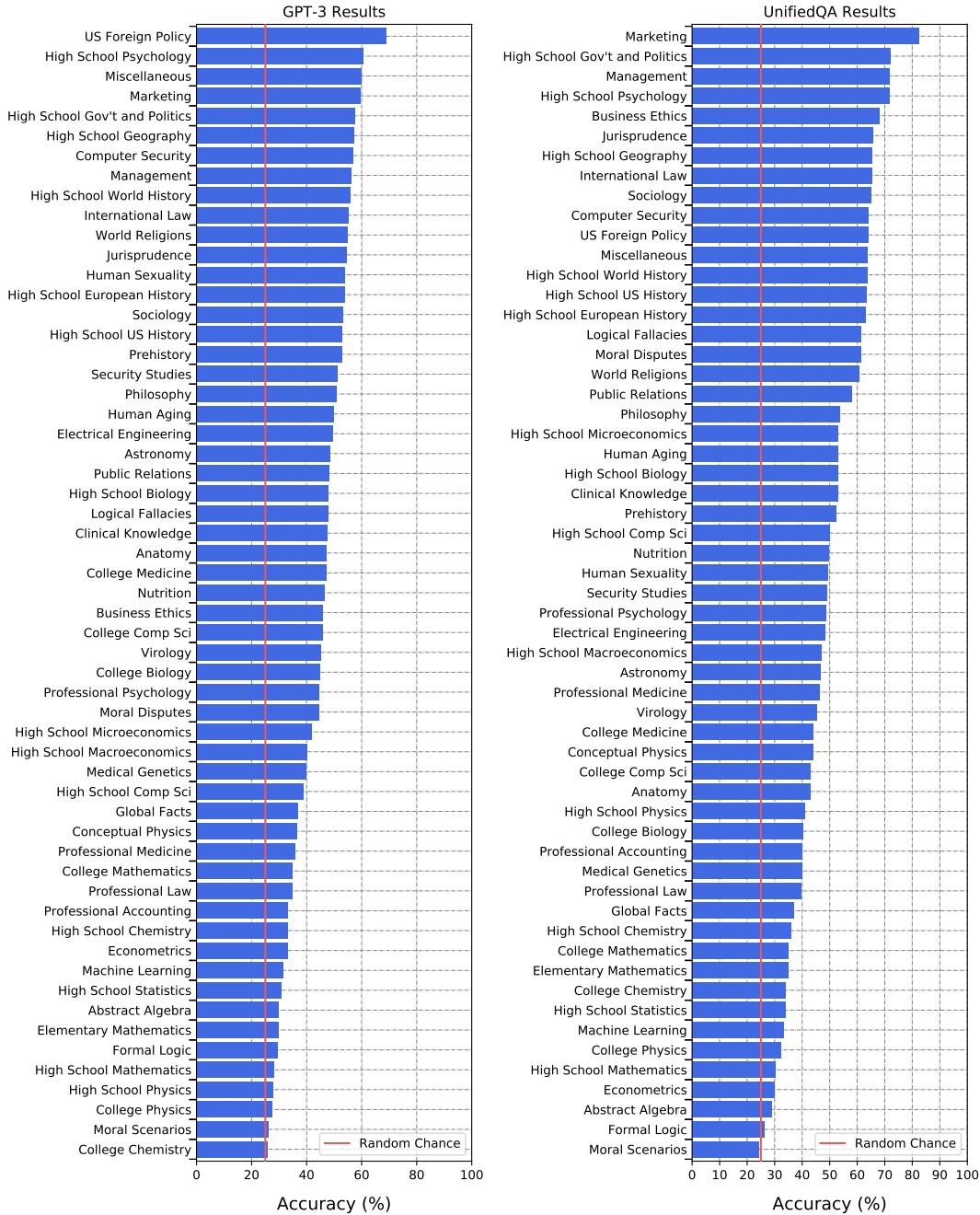

Figure 9: On the left are GPT-3 few shot accuracies for all of the 57 tasks. On the right are UnifiedQA transfer accuracies for all of the 57 tasks. For both models, capabilities are lopsided.

## A.1    ANALYSIS WITH MORE FINE-TUNED MODELS

We primarily analyzed models with more than 10 billion parameters in the main body of the paper. For this section, we analyze smaller models including RoBERTa-base (125 million parameters) (Liu

et al., 2019), ALBERT-xxlarge (223 million parameters) (Lan et al., 2020), and GPT-2 (1,558 million parameters) (Radford et al., 2019). Models are fine-tuned to predict one of four classes using the UnifiedQA MCQ questions and using our dev+val set. We test on our multitask test set.

We observe that these smaller models can attain better-than-random accuracy. RoBERTa-base attains an overall accuracy of 27.9%, with 27.9% accuracy for the humanities, 28.8% for social sciences, 27.0% for STEM, and 27.7% for other. ALBERT-xxlarge attains an accuracy of 27.1%, with 27.2% accuracy for the humanities, 25.7% for the social sciences, 27.7% for STEM, and 27.9% for other. GPT-2 attains an accuracy of 32.4%, with 32.8% accuracy for the humanities, 33.3% for the social sciences, 30.2% for STEM, and 33.1% for other.

Compare this to UnifiedQA's smallest variant, which has just 60 million parameters and approximately 29.3% accuracy. It obtains higher accuracy than RoBERTa and ALBERT, even though it has fewer parameters. This suggests that its larger pretraining dataset enables higher accuracy. Likewise, UnifiedQA with 3 billion parameters attains 43.7%, while the similarly sized GPT-2 model with 1.5 billion parameters attains 32.4% accuracy. This again suggests that T5's larger pretraining dataset size (and therefore UnifiedQA's pretraining dataset size) can increase accuracy.

## A.2 Error Analysis

We qualitatively analyze when GPT-3 makes high confidence mistakes. We find that while many of these mistakes were clearly wrong, many were mistakes that a human might make. For example, one question it got wrong was "How many chromosomes do all human somatic cells contain?" The correct answer is 46, while few-shot GPT-3 predicted 23 with confidence 97.5%. This answer would have been correct if the question asked about the number of *pairs* of chromosomes. Similarly, many of its other high confidence mistakes were also correct answers to slightly different questions.

## A.3 Format Sensitivity

While different question formatting choices often lead to similar GPT-3 accuracies, we find that UnifiedQA is more sensitive. UnifiedQA's input format is of the form

```
QUESTION1 \\n (A) CHOICE1 (B) CHOICE2 (C) CHOICE3 (D) CHOICE4
```

where questions and choices are normalized and made lowercase. If we remove the `` from the input, accuracy declines by several percentage points.

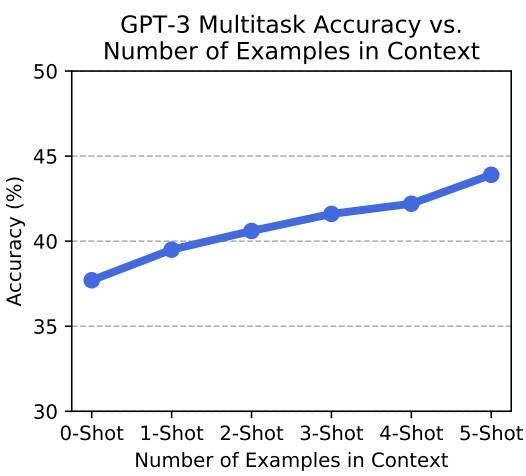

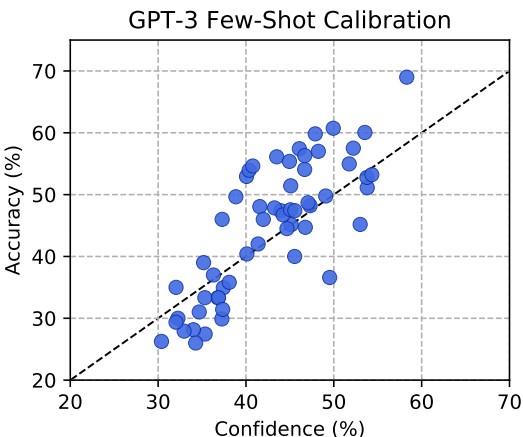

Figure 10: As the number of few-shot instruction examples increases, the accuracy monotonically increases. Notably, zero-shot performance is only somewhat lower than 5-shot accuracy.

Figure 11: While models are more calibrated in a few-shot setting than a zero-shot setting, they are still miscalibrated, with gap between accuracy and confidence reaching up to $14\%$. Here the correlation between confidence and accuracy is $r = 0.81$, compared to $r = 0.63$ in the zero-shot setting.

## B  TEST DETAILS

### B.1  TASK DESCRIPTIONS AND EXAMPLES

We provide analysis of question length and difficulty in Figure 12. We list all tasks and the topics they test in Table 2. We also provide an example for each task starting with Figure 14.

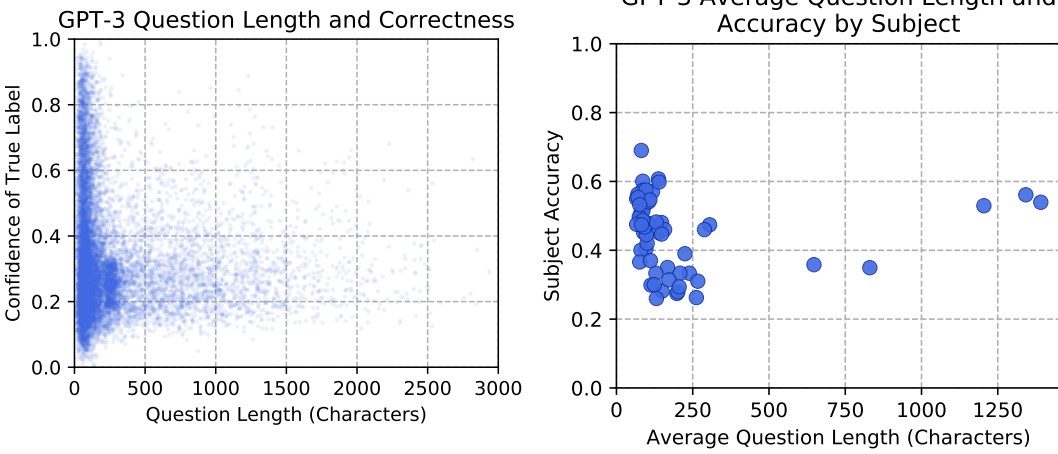

Figure 12: Figures on the relation between question difficulty and question length. For questions longer than a tweet (280 characters), the correlation between question length and true label confidence is slightly positive. This shows that longer questions are not necessarily harder.

### B.2  EXACT QUESTION AND ANSWER CONTAMINATION

Since language models train on vast text corpora, there is some chance that they have seen the exact question and answer during pretraining. If they memorized the exact question and answer, then they would attain higher accuracy than their true ability. Likewise, a question's entropy would be especially low if it were memorized. Memorized questions and answers should have low entropy and

high accuracy. However, in Figure 13, we see that accuracy and question entropy are not positively correlated, suggesting that the test's low-entropy questions do not correspond to memorized (and thereby correctly predicted) answers. This suggests that our *exact* questions were not memorized. However, during pretraining models encountered text *related* to our questions through processing Wikipedia. We also note that most of our questions came from PDFs or websites where questions and answers are on separate pages.

See Brown et al. (2020) for a previous discussion of contamination showing that the phenomena hardly affects performance. To reduce the probability that future models encounter exact questions during test-time, we will provide a list of question sources.

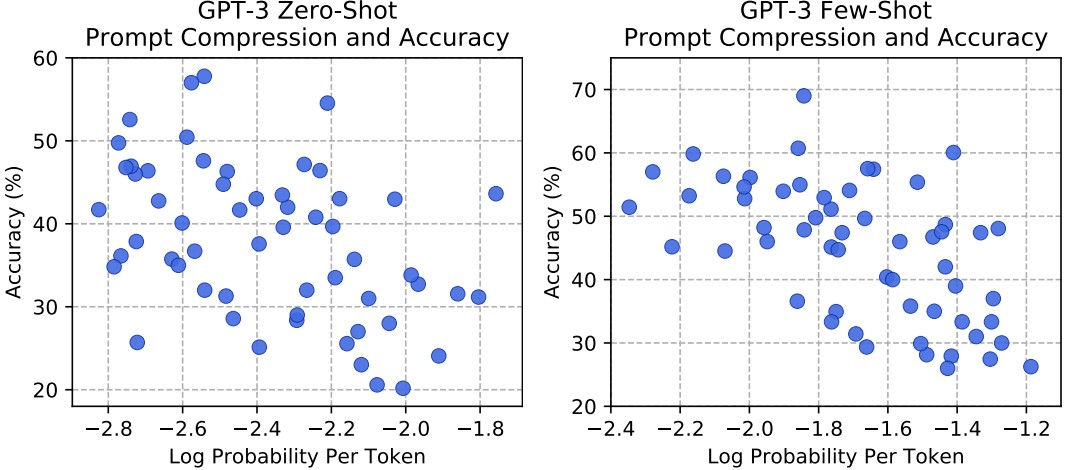

Figure 13: The average log probability of the question (without answer) is not strongly positively correlated with accuracy, all else equal. Each point corresponds to a task. Higher log probability indicates higher compression, and especially high log probability would suggest memorization. In the zero-shot question prompt, the correlation between average log probability and accuracy is $r = -0.43$, and for the few-shot setting the correlation is $r = -0.56$.

| Task | Tested Concepts | Supercategory |
|---|---|---|
| Abstract Algebra | Groups, rings, fields, vector spaces, ... | STEM |
| Anatomy | Central nervous system, circulatory system, ... | STEM |
| Astronomy | Solar system, galaxies, asteroids, ... | STEM |
| Business Ethics | Corporate responsibility, stakeholders, regulation, ... | Other |
| Clinical Knowledge | Spot diagnosis, joints, abdominal examination, ... | Other |
| College Biology | Cellular structure, molecular biology, ecology, ... | STEM |
| College Chemistry | Analytical, organic, inorganic, physical, ... | STEM |
| College Computer Science | Algorithms, systems, graphs, recursion, ... | STEM |
| College Mathematics | Differential equations, real analysis, combinatorics, ... | STEM |
| College Medicine | Introductory biochemistry, sociology, reasoning, ... | Other |
| College Physics | Electromagnetism, thermodynamics, special relativity, ... | STEM |
| Computer Security | Cryptography, malware, side channels, fuzzing, ... | STEM |
| Conceptual Physics | Newton's laws, rotational motion, gravity, sound, ... | STEM |
| Econometrics | Volatility, long-run relationships, forecasting, ... | Social Sciences |
| Electrical Engineering | Circuits, power systems, electrical drives, ... | STEM |
| Elementary Mathematics | Word problems, multiplication, remainders, rounding, ... | STEM |
| Formal Logic | Propositions, predicate logic, first-order logic, ... | Humanities |
| Global Facts | Extreme poverty, literacy rates, life expectancy, ... | Other |
| High School Biology | Natural selection, heredity, cell cycle, Krebs cycle, ... | STEM |
| High School Chemistry | Chemical reactions, ions, acids and bases, ... | STEM |
| High School Computer Science | Arrays, conditionals, iteration, inheritance, ... | STEM |
| High School European History | Renaissance, reformation, industrialization, ... | Humanities |
| High School Geography | Population migration, rural land-use, urban processes, ... | Social Sciences |
| High School Gov't and Politics | Branches of government, civil liberties, political ideologies, ... | Social Sciences |
| High School Macroeconomics | Economic indicators, national income, international trade, ... | Social Sciences |
| High School Mathematics | Pre-algebra, algebra, trigonometry, calculus, ... | STEM |
| High School Microeconomics | Supply and demand, imperfect competition, market failure, ... | Social Sciences |
| High School Physics | Kinematics, energy, torque, fluid pressure, ... | STEM |
| High School Psychology | Behavior, personality, emotions, learning, ... | Social Sciences |
| High School Statistics | Random variables, sampling distributions, chi-square tests, ... | STEM |
| High School US History | Civil War, the Great Depression, The Great Society, ... | Humanities |
| High School World History | Ottoman empire, economic imperialism, World War I, ... | Humanities |
| Human Aging | Senescence, dementia, longevity, personality changes, ... | Other |
| Human Sexuality | Pregnancy, sexual differentiation, sexual orientation, ... | Social Sciences |
| International Law | Human rights, sovereignty, law of the sea, use of force, ... | Humanities |
| Jurisprudence | Natural law, classical legal positivism, legal realism, ... | Humanities |
| Logical Fallacies | No true Scotsman, base rate fallacy, composition fallacy, ... | Humanities |
| Machine Learning | SVMs, VC dimension, deep learning architectures, ... | STEM |
| Management | Organizing, communication, organizational structure, ... | Other |
| Marketing | Segmentation, pricing, market research, ... | Other |
| Medical Genetics | Genes and cancer, common chromosome disorders, ... | Other |
| Miscellaneous | Agriculture, Fermi estimation, pop culture, ... | Other |
| Moral Disputes | Freedom of speech, addiction, the death penalty, ... | Humanities |
| Moral Scenarios | Detecting physical violence, stealing, externalities, ... | Humanities |
| Nutrition | Metabolism, water-soluble vitamins, diabetes, ... | Other |
| Philosophy | Skepticism, phronesis, skepticism, Singer's Drowning Child, ... | Humanities |
| Prehistory | Neanderthals, Mesoamerica, extinction, stone tools, ... | Humanities |
| Professional Accounting | Auditing, reporting, regulation, valuation, ... | Other |
| Professional Law | Torts, criminal law, contracts, property, evidence, ... | Humanities |
| Professional Medicine | Diagnosis, pharmacotherapy, disease prevention, ... | Other |
| Professional Psychology | Diagnosis, biology and behavior, lifespan development, ... | Social Sciences |
| Public Relations | Media theory, crisis management, intelligence gathering, ... | Social Sciences |
| Security Studies | Environmental security, terrorism, weapons of mass destruction, ... | Social Sciences |
| Sociology | Socialization, cities and community, inequality and wealth, ... | Social Sciences |
| US Foreign Policy | Soft power, Cold War foreign policy, isolationism, ... | Social Sciences |
| Virology | Epidemiology, coronaviruses, retroviruses, herpesviruses, ... | Other |
| World Religions | Judaism, Christianity, Islam, Buddhism, Jainism, ... | Humanities |

Table 2: Summary of all 57 tasks.

Find all $c$ in $\mathbb{Z}_3$ such that $\mathbb{Z}_3[x]/(x^2 + c)$ is a field.
(A) 0 **(B) 1** (C) 2 (D) 3

Figure 14: An Abstract Algebra example.

What is the embryological origin of the hyoid bone?
(A) The first pharyngeal arch
(B) The first and second pharyngeal arches
(C) The second pharyngeal arch
**(D) The second and third pharyngeal arches**

Figure 15: An Anatomy example.

Why isn't there a planet where the asteroid belt is located?
(A) A planet once formed here but it was broken apart by a catastrophic collision.
(B) There was not enough material in this part of the solar nebula to form a planet.
(C) There was too much rocky material to form a terrestrial planet but not enough gaseous material to form a jovian planet.
**(D) Resonance with Jupiter prevented material from collecting together to form a planet.**

Figure 16: An Astronomy example.

Three contrasting tactics that CSO's can engage in to meet their aims are _________ which typically involves research and communication, _________, which may involve physically attacking a company's operations or _________, often involving some form of _________.
(A) Non-violent direct action, Violent direct action, Indirect action, Boycott
(B) Indirect action, Instrumental action, Non-violent direct action, Information campaign
**(C) Indirect action, Violent direct action, Non-violent direct-action Boycott.**
(D) Non-violent direct action, Instrumental action, Indirect action, Information campaign

Figure 17: A Business Ethics example.

How many attempts should you make to cannulate a patient before passing the job on to a senior colleague?
(A) 4 (B) 3 **(C) 2** (D) 1

Figure 18: A Clinical Knowledge example.

In a given population, 1 out of every 400 people has a cancer caused by a completely recessive allele, b. Assuming the population is in Hardy-Weinberg equilibrium, which of the following is the expected proportion of individuals who carry the b allele but are not expected to develop the cancer?
(A) 1/400 (B) 19/400 (C) 20/400 **(D) 38/400**

Figure 19: A College Biology example.

Which of the following statements about the lanthanide elements is NOT true?
(A) The most common oxidation state for the lanthanide elements is +3.
(B) Lanthanide complexes often have high coordination numbers (> 6).
(C) All of the lanthanide elements react with aqueous acid to liberate hydrogen.
**(D) The atomic radii of the lanthanide elements increase across the period from La to Lu.**

Figure 20: A College Chemistry example.

Consider a computer design in which multiple processors, each with a private cache memory, share global memory using a single bus. This bus is the critical system resource. Each processor can execute one instruction every 500 nanoseconds as long as memory references are satisfied by its local cache. When a cache miss occurs, the processor is delayed for an additional 2,000 nanoseconds. During half of this additional delay, the bus is dedicated to serving the cache miss. During the other half, the processor cannot continue, but the bus is free to service requests from other processors. On average, each instruction requires 2 memory references. On average, cache misses occur on 1 percent of references. What proportion of the capacity of the bus would a single processor consume, ignoring delays due to competition from other processors?
(A) 1/50    **(B) 1/27**    (C) 1/25    (D) 2/27

Figure 21: A College Computer Science example.

Let $A$ be a real $2 \times 2$ matrix. Which of the following statements must be true?
I. All of the entries of $A^2$ are nonnegative.
II. The determinant of $A^2$ is nonnegative.
III. If A has two distinct eigenvalues, then $A^2$ has two distinct eigenvalues.
(A) I only    **(B) II only**    (C) III only    (D) II and III only

Figure 22: A College Mathematics example.

In a genetic test of a newborn, a rare genetic disorder is found that has X-linked recessive transmission. Which of the following statements is likely true regarding the pedigree of this disorder?
(A) All descendants on the maternal side will have the disorder.
(B) Females will be approximately twice as affected as males in this family.
**(C) All daughters of an affected male will be affected.**
(D) There will be equal distribution of males and females affected.

Figure 23: A College Medicine example.

One end of a Nichrome wire of length 2L and cross-sectional area A is attached to an end of another Nichrome wire of length L and cross- sectional area 2A. If the free end of the longer wire is at an electric potential of 8.0 volts, and the free end of the shorter wire is at an electric potential of 1.0 volt, the potential at the junction of the two wires is most nearly equal to
**(A) 2.4 V**
(B) 3.3 V
(C) 4.5 V
(D) 5.7 V

Figure 24: A College Physics example.

Why is it that anti-virus scanners would not have found an exploitation of Heartbleed?
(A) It's a vacuous question: Heartbleed only reads outside a buffer, so there is no possible exploit
(B) Anti-virus scanners tend to look for viruses and other malicious
(C) Heartbleed attacks the anti-virus scanner itself
**(D) Anti-virus scanners tend to look for viruses and other malicious code, but Heartbleed exploits steal secrets without injecting any code**

Figure 25: A Computer Security example.

A model airplane flies slower when flying into the wind and faster with wind at its back. When launched at right angles to the wind, a cross wind, its groundspeed compared with flying in still air is
(A) the same    **(B) greater**    (C) less    (D) either greater or less depending on wind speed

Figure 26: A Conceptual Physics example.

Consider the following AR(1) model with the disturbances having zero mean and unit variance
$y_t = 0.2 + 0.4y_{t-1} + u_t$
The (unconditional) mean of $y$ will be given by
(A) 0.2    (B) 0.4    (C) 0.5    **(D) 0.33**

Figure 27: An Econometrics example.

A point pole has a strength of $4\pi \times 10^{-4}$ weber. The force in newtons on a point pole of $4\pi \times 1.5 \times 10^{-4}$ weber placed at a distance of 10 cm from it will be
**(A) 15 N.**    (B) 20 N.    (C) 7.5 N.    (D) 3.75 N.

Figure 28: An Electrical Engineering example.

A total of 30 players will play basketball at a park. There will be exactly 5 players on each team. Which statement correctly explains how to find the number of teams needed?
(A) Add 5 to 30 to find 35 teams.
**(B) Divide 30 by 5 to find 6 teams.**
(C) Multiply 30 and 5 to find 150 teams.
(D) Subtract 5 from 30 to find 25 teams.

Figure 29: An Elementary Mathematics example.

Determine whether the statements are logically equivalent or contradictory. If neither, determine whether they are consistent or inconsistent.
$E \supset (F \cdot E)$ and $\sim E \cdot F$
(A) Logically equivalent
(B) Contradictory
**(C) Neither logically equivalent nor contradictory, but consistent**
(D) Inconsistent

Figure 30: A Formal Logic example.

As of 2017, how many of the world's 1-year-old children today have been vaccinated against some disease?
**(A) 80%**
(B) 60%
(C) 40%
(D) 20%

Figure 31: A Global Facts example.

Homologous structures are often cited as evidence for the process of natural selection. All of the following are examples of homologous structures EXCEPT
(A) the wings of a bird and the wings of a bat
(B) the flippers of a whale and the arms of a man
(C) the pectoral fins of a porpoise and the flippers of a seal
**(D) the forelegs of an insect and the forelimbs of a dog**

Figure 32: A High School Biology example.

From the solubility rules, which of the following is true?
(A) All chlorides, bromides, and iodides are soluble
(B) All sulfates are soluble
(C) All hydroxides are soluble
**(D) All ammonium-containing compounds are soluble**

Figure 33: A High School Chemistry example.

A list of numbers has n elements, indexed from 1 to n. The following algorithm is intended to display the number of elements in the list that have a value greater than 100. The algorithm uses the variables count and position. Steps 3 and 4 are missing.
Step 1: Set count to 0 and position to 1.
Step 2: If the value of the element at index position is greater
than 100, increase the value of count by 1.
Step 3: (missing step)
Step 4: (missing step)
Step 5: Display the value of count.
Which of the following could be used to replace steps 3 and 4 so that the algorithm works as intended?
(A) Step 3: Increase the value of position by 1.
Step 4: Repeat steps 2 and 3 until the value of count is greater than 100.
(B) Step 3: Increase the value of position by 1.
Step 4: Repeat steps 2 and 3 until t he value of position is greater than n.
(C) Step 3: Repeat step 2 until the value of count is greater than 100.
Step 4: Increase the value of position by 1.
**(D) Step 3: Repeat step 2 until the value of position is greater than n.**
**Step 4: Increase the value of count by 1.**

Figure 34: A High School Computer Science example.

This question refers to the following information.
Albeit the king's Majesty justly and rightfully is and ought to be the supreme head of the Church of England, and so is recognized by the clergy of this realm in their convocations, yet nevertheless, for corroboration and confirmation thereof, and for increase of virtue in Christ's religion within this realm of England, and to repress and extirpate all errors, heresies, and other enormities and abuses heretofore used in the same, be it enacted, by authority of this present Parliament, that the king, our sovereign lord, his heirs and successors, kings of this realm, shall be taken, accepted, and reputed the only supreme head in earth of the Church of England, called Anglicans Ecclesia; and shall have and enjoy, annexed and united to the imperial crown of this realm, as well the title and style thereof, as all honors, dignities, preeminences, jurisdictions, privileges, authorities, immunities, profits, and commodities to the said dignity of the supreme head of the same Church belonging and appertaining; and that our said sovereign lord, his heirs and successors, kings of this realm, shall have full power and authority from time to time to visit, repress, redress, record, order, correct, restrain, and amend all such errors, heresies, abuses, offenses, contempts, and enormities, whatsoever they be, which by any manner of spiritual authority or jurisdiction ought or may lawfully be reformed, repressed, ordered, redressed, corrected, restrained, or amended, most to the pleasure of Almighty God, the increase of virtue in Christ's religion, and for the conservation of the peace, unity, and tranquility of this realm; any usage, foreign land, foreign authority, prescription, or any other thing or things to the contrary hereof notwithstanding.
English Parliament, Act of Supremacy, 1534
From the passage, one may infer that the English Parliament wished to argue that the Act of Supremacy would
(A) give the English king a new position of authority
(B) give the position of head of the Church of England to Henry VIII alone and exclude his heirs
(C) establish Calvinism as the one true theology in England
**(D) end various forms of corruption plaguing the Church in England**

Figure 35: A High School European History example.

During the third stage of the demographic transition model, which of the following is true?
(A) Birth rates increase and population growth rate is less rapid.
**(B) Birth rates decline and population growth rate is less rapid.**
(C) Birth rates increase and population growth rate increases.
(D) Birth rates decrease and population growth rate increases.

Figure 36: A High School Geography example.

Which of the following best states an argument made by James Madison in The Federalist number 10?
(A) Honest politicians can prevent factions from developing.
(B) Factions are more likely to occur in large republics than in small ones.
**(C) The negative effects of factionalism can be reduced by a republican government.**
(D) Free elections are the people's best defense against factionalism.

Figure 37: A High School Government and Politics example.

Which of the following is not included in the U.S. GDP?
(A) The U.S. military opens a new base in a foreign country with 1000 U.S. personnel.
(B) Japanese consumers buy thousands of CDs produced in the United States.
**(C) An American pop singer performs a sold-out concert in Paris.**
(D) A French theatrical production tours dozens of American cities.

Figure 38: A High School Macroeconomics example.

Joe was in charge of lights for a dance. The red light blinks every two seconds, the yellow light every three seconds, and the blue light every five seconds. If we include the very beginning and very end of the dance, how many times during a seven minute dance will all the lights come on at the same time? (Assume that all three lights blink simultaneously at the very beginning of the dance.)
(A) 3
**(B) 15**
(C) 6
(D) 5

Figure 39: A High School Mathematics example.

If the government subsidizes producers in a perfectly competitive market, then
(A) the demand for the product will increase
(B) the demand for the product will decrease
**(C) the consumer surplus will increase**
(D) the consumer surplus will decrease

Figure 40: A High School Microeconomics example.

A point charge, $Q = +1$ mC, is fixed at the origin. How much work is required to move a charge, $Q = +8$ μC, from the point (0, 4 meters) to the point (3 meters, 0)?
(A) 3.5 J
**(B) 6.0 J**
(C) 22.5 J
(D) 40 J

Figure 41: A High School Physics example.

While swimming in the ocean, Ivan is frightened by a dark shadow in the water even before he has the chance to identify what the shadow is. The synaptic connections taking place during this incident of fright are best described by which of the following?
**(A) Messages are sent from the thalamus directly to the amygdala.**
(B) Messages are sent from the thalamus to the "what" and "where" pathways.
(C) Messages are sent from the parasympathetic nervous system to the cerebral cortex.
(D) Messages are sent from the frontal lobes to the pituitary gland.

Figure 42: A High School Psychology example.

Jonathan obtained a score of 80 on a statistics exam, placing him at the 90th percentile. Suppose five points are added to everyone's score. Jonathan's new score will be at the
(A) 80th percentile.
(B) 85th percentile.
**(C) 90th percentile.**
(D) 95th percentile.

Figure 43: A High School Statistics example.

This question refers to the following information.
"Society in every state is a blessing, but government even in its best state is but a necessary evil; in its worst state an intolerable one; for when we suffer, or are exposed to the same miseries by a government, which we might expect in a country without government, our calamity is heightened by reflecting that we furnish the means by which we suffer. Government, like dress, is the badge of lost innocence; the palaces of kings are built on the ruins of the bowers of paradise. For were the impulses of conscience clear, uniform, and irresistibly obeyed, man would need no other lawgiver; but that not being the case, he finds it necessary to surrender up a part of his property to furnish means for the protection of the rest; and this he is induced to do by the same prudence which in every other case advises him out of two evils to choose the least. Wherefore, security being the true design and end of government, it unanswerably follows that whatever form thereof appears most likely to ensure it to us, with the least expense and greatest benefit, is preferable to all others."
Thomas Paine, Common Sense, 1776
Which of the following "miseries" alluded to above were most condemned by Anti-Federalists of the post-Revolutionary era?
(A) Organized response to Bacon's Rebellion.
(B) Federal response to Shays's Rebellion.
**(C) Federal response to the Whiskey Rebellion.**
(D) Federal response to Pontiac's Rebellion.

Figure 44: A High School US History example.

This question refers to the following information.
"The real grievance of the worker is the insecurity of his existence; he is not sure that he will always have work, he is not sure that he will always be healthy, and he foresees that he will one day be old and unfit to work. If he falls into poverty, even if only through a prolonged illness, he is then completely helpless, left to his own devices, and society does not currently recognize any real obligation towards him beyond the usual help for the poor, even if he has been working all the time ever so faithfully and diligently. The usual help for the poor, however, leaves a lot to be desired, especially in large cities, where it is very much worse than in the country."
Otto von Bismarck, 1884
Otto von Bismarck likely made this speech in reaction to which of the following issues?
(A) Social acceptance of child labor.
(B) Declining life expectancy in Germany.
**(C) Criticisms of German trade tariffs.**
(D) Negative effects attributed to industrial capitalism.

Figure 45: A High School World History example.

All other things being equal, which of the following persons is more likely to show osteoporosis?
(A) An older Hispanic American woman
(B) An older African American woman
**(C) An older Asian American woman**
(D) An older Native American woman

Figure 46: A Human Aging example.

Morning sickness is typically a problem:
**(A) during the first trimester**
(B) during the second trimester
(C) during the third trimester
(D) all through the pregnancy

Figure 47: A Human Sexuality example.

Would a reservation to the definition of torture in the ICCPR be acceptable in contemporary practice?
(A) This is an acceptable reservation if the reserving country's legislation employs a different definition
**(B) This is an unacceptable reservation because it contravenes the object and purpose of the ICCPR**
(C) This is an unacceptable reservation because the definition of torture in the ICCPR is consistent with customary international law
(D) This is an acceptable reservation because under general international law States have the right to enter reservations to treaties

Figure 48: An International Law example.

Which position does Rawls claim is the least likely to be adopted by the POP (people in the original position)?
**(A) The POP would choose equality above liberty.**
(B) The POP would opt for the 'maximin' strategy.
(C) The POP would opt for the 'difference principle.'
(D) The POP would reject the 'system of natural liberty.'

Figure 49: A Jurisprudence example.

John Stuart Mill: Each person's happiness is a good to that person, and the general happiness, therefore, a good to the aggregate of all persons.
**(A) Fallacy of Composition**
(B) Fallacy of Division
(C) Gambler's Fallacy
(D) Equivocation

Figure 50: A Logical Fallacies example.

A 6-sided die is rolled 15 times and the results are: side 1 comes up 0 times; side 2: 1 time; side 3: 2 times; side 4: 3 times; side 5: 4 times; side 6: 5 times. Based on these results, what is the probability of side 3 coming up when using Add-1 Smoothing?
(A) 2/15    **(B) 1/7**    (C) 3/16    (D) 1/5

Figure 51: A Machine Learning example.

According to Lewin, Lippet and White's 1939 experiment, which form of leadership produced the most work from participants?
(A) Laissez-faire
(B) Democratic
**(C) Authoritarian**
(D) A mix of laissez-faire and democratic

Figure 52: A Management example.

The single group within society that is most vulnerable to reference group influence is:
(A) The older consumer who feels somewhat left out of things.
(B) The married women, many of whom feel a need for stability in their lives.
(C) New immigrants who really want to assimilate into their new culture.
**(D) Children, who base most of their buying decisions on outside influences.**

Figure 53: A Marketing example.

Which of the following conditions does not show multifactorial inheritance?
(A) Pyloric stenosis
(B) Schizophrenia
(C) Spina bifida (neural tube defects)
**(D) Marfan syndrome**

Figure 54: A Medical Genetics example.

If each of the following meals provides the same number of calories, which meal requires the most land to produce the food?
(A) Red beans and rice
**(B) Steak and a baked potato**
(C) Corn tortilla and refried beans
(D) Lentil soup and brown bread

Figure 55: A Miscellaneous example.

In response to Sandel's "social justice" argument, Kamm argues that
(A) even if we were able to enhance ourselves or others, we would not thereby be obligated to do so.
(B) there is a difference between (i) choosing that one have a certain trait and (ii) being responsible for bearing the costs of that choice.
(C) having the option to enhance will lead to fewer instances of people who require the assistance of others.
**(D) all of the above**

Figure 56: A Moral Disputes example.

For which of these two scenarios does the main character (who uses I/me/my) do something clearly morally wrong?
Scenario 1 | I drive very fast when I'm on a racetrack.
Scenario 2 | I drive very fast when I'm in the suburbs.
(A) Wrong, Wrong
(B) Wrong, Not wrong
**(C) Not wrong, Wrong**
(D) Not wrong, Not wrong

Figure 57: A Moral Scenarios example. The formatting of this task hinders UnifiedQA performance substantially.

Which of the following is the most plausible explanation for the protective effect of dietary fibre against cancer of the colon?
(A) Propionic acid, formed during colonic fibre fermentation inhibits liver fatty acid synthesis
(B) Butyric acid, formed during colonic fibre fermentation stimulates "silencing" of the SLC5A8 tumour suppressor gene
**(C) Butyric acid, formed during colonic fibre fermentation stimulates anti-oxidant defences in the colon**
(D) None of these options are correct

Figure 58: A Nutrition example.

According to Moore's "ideal utilitarianism," the right action is the one that brings about the greatest amount of:
(A) pleasure.
(B) happiness.
**(C) good.**
(D) virtue.

Figure 59: A Philosophy example.

Researchers now believe that the decline of the Maya was caused chiefly by:
(A) a cataclysm of some kind, such as an earthquake, volcano, or tsunami.
**(B) ecological degradation resulting from slash-and-burn farming techniques.**
(C) endless wars between neighboring Mayan city-states.
(D) practices of interbreeding that led to a steep rise in congenital disorders.

Figure 60: A Prehistory example.

Krete is an unmarried taxpayer with income exclusively from wages. By December 31, year 1, Krete's employer has withheld $16,000 in federal income taxes and Krete has made no estimated tax payments. On April 15, year 2, Krete timely filed for an extension request to file her individual tax return, and paid $300 of additional taxes. Krete's year 1 tax liability was $16,500 when she timely filed her return on April 30, year 2, and paid the remaining tax liability balance. What amount would be subject to the penalty for underpayment of estimated taxes?
**(A) $0**
(B) $500
(C) $1,650
(D) $16,500

Figure 61: A Professional Accounting example.

The night before his bar examination, the examinee's next-door neighbor was having a party. The music from the neighbor's home was so loud that the examinee couldn't fall asleep. The examinee called the neighbor and asked her to please keep the noise down. The neighbor then abruptly hung up. Angered, the examinee went into his closet and got a gun. He went outside and fired a bullet through the neighbor's living room window. Not intending to shoot anyone, the examinee fired his gun at such an angle that the bullet would hit the ceiling. He merely wanted to cause some damage to the neighbor's home to relieve his angry rage. The bullet, however, ricocheted off the ceiling and struck a partygoer in the back, killing him. The jurisdiction makes it a misdemeanor to discharge a firearm in public. The examinee will most likely be found guilty for which of the following crimes in connection to the death of the partygoer?
**(A) Murder.**
(B) Involuntary manslaughter.
(C) Voluntary manslaughter.
(D) Discharge of a firearm in public.

Figure 62: A Professional Law example.

A 63-year-old man is brought to the emergency department because of a 4-day history of increasingly severe left leg pain and swelling of his left calf. He also has a 1-month history of increasingly severe upper midthoracic back pain. During this time, he has had a 9-kg (20-lb) weight loss despite no change in appetite. He has no history of major medical illness. His only medication is ibuprofen. He is 180 cm (5 ft 11 in) tall and weighs 82 kg (180 lb); BMI is 25 kg/m2 . His vital signs are within normal limits. On examination, lower extremity pulses are palpable bilaterally. The remainder of the physical examination shows no abnormalities. An x-ray of the thoracic spine shows no abnormalities. A CT scan of the abdomen shows a 3-cm mass in the body of the pancreas; there are liver metastases and encasement of the superior mesenteric artery. Ultrasonography of the left lower extremity shows a femoropopliteal venous clot. Which of the following is the most likely cause of this patient's symptoms?
(A) Carcinoid syndrome
**(B) Hypercoagulability from advanced malignancy**
(C) Multiple endocrine neoplasia
(D) Splenic artery aneurysm and embolic disease of the left lower extremity

Figure 63: A Professional Medicine example.

The technique that is most likely to produce an immediate improvement in the behavior of a child who hits others and rips up schoolbooks is
**(A) a combination of reinforcement for appropriate behavior and mild punishment for inappropriate behavior**
(B) differential positive reinforcement
(C) loss of privileges for misbehaving
(D) shaping

Figure 64: A Professional Psychology example.

You work for a utility company that is building a biomass plant in the community. Your employer asks you to give a series of community talks about the plant and future operations. You visit the plant several hours before you are due to give a speech that has been prepared by your immediate supervisor. During the tour of the plant, you discover several claims in the speech are not true. What do you do?
(A) Write and deliver a new speech that you know is entirely correct.
(B) Cancel all speeches until you and your supervisor can get the information straight.
(C) Deliver the speech as prepared and discuss the inaccuracies with your supervisor afterward.
**(D) Address the inaccuracies with your supervisor immediately and make the necessary corrections before giving the speech.**

Figure 65: A Public Relations example.

The Chemical Weapons Convention (CWC) prohibited the possession or deployment of chemical weapons; however it failed to implement stipulations that would require signatories to declare their existing stocks of chemical weapons, to identify facilities that were once involved in chemical production, or to announce when their existing stocks would be destroyed.
(A) The Chemical Weapons Convention (CWC) prohibited the possession or deployment of chemical weapons; however it failed to implement stipulations that would require signatories to declare their existing stocks of chemical weapons, to identify facilities that were once involved in chemical production, or to announce when their existing stocks would be destroyed.
(B) The CWC made some important developments regarding the use and possession of chemical weapons and the destruction of existing stockpiles. However, the treaty failed to establish an independent body empowered with the capacity to check treaty compliance. Lack of supra-state authority has undermined the ability to enforce those developments. Given the anarchical nature of international society it may be in the national security interest to retain stocks.
(C) Chemical weapons continue to exert a determining influence on international society. As early as the 1970s military strategists were convinced of the deterrence effects chemical weapons could have, comparable to the second strike survival logic of nuclear deterrence. The preferences of strategists resulted in continued manufacture and stockpiling of weapons creating an international crisis of stability.
**(D) While the CWC has been ratified by the majority of international society, some nations with a large chemical capability at their disposal have yet to enter into the treaty. However, to some analysts the destructive military potential would be limited, having a moderate effect on a well-equipped army in conventional warfare. Chemical arsenal essentially falls under the category of the "poor mans" weaponry, being simplistic and inexpensive whilst having limited military utility. However, the concern remains of the prospective impact a terrorist chemical attack could have on civilian populations.**

Figure 66: A Security Studies example.

Which of the following statements most closely corresponds with differential association theory?
**(A) If all of your friends jumped off a bridge, I suppose you would too.**
(B) You should be proud to be a part of this organization.
(C) If the door is closed, try the window.
(D) Once a thief, always a thief.

Figure 67: A Sociology example.

Why did Congress oppose Wilson's proposal for the League of Nations?
(A) It feared the League would encourage Soviet influence in the US
(B) It feared the League would be anti-democratic
**(C) It feared the League would commit the US to an international alliance**
(D) Both a and b

Figure 68: A US Foreign Policy example.

An observational study in diabetics assesses the role of an increased plasma fibrinogen level on the risk of cardiac events. 130 diabetic patients are followed for 5 years to assess the development of acute coronary syndrome. In the group of 60 patients with a normal baseline plasma fibrinogen level, 20 develop acute coronary syndrome and 40 do not. In the group of 70 patients with a high baseline plasma fibrinogen level, 40 develop acute coronary syndrome and 30 do not. Which of the following is the best estimate of relative risk in patients with a high baseline plasma fibrinogen level compared to patients with a normal baseline plasma fibrinogen level?
(A) (40/30)/(20/40)
(B) (40*40)/(20*30)
**(C) (40*70)/(20*60)**
(D) (40/70)/(20/60)

Figure 69: A Virology example.

The Great Cloud Sutra prophesied the imminent arrival of which person?
**(A) Maitreya (Milo)**
(B) The Buddha
(C) Zhou Dunyi
(D) Wang Yangming

Figure 70: A World Religions example.

