# OpenReview forum: "Measuring Massive Multitask Language Understanding"
_ICLR.cc/2021/Conference — ICLR 2021 Poster_

### Official Review · AnonReviewer1 · 2020-10-27
**This paper focuses on coming up with 57 different tasks and measure the performance of these large scale transformer models such as GPT3 on these different tasks.**

**Rating:** 6
**Confidence:** 4

**Review:**

This paper focuses on coming up with 57 different tasks and measure the performance of these large scale transformer models such as GPT3 on these different tasks. The main claims of this paper are to demonstrate these large-scale models still struggle to use the knowledge it has learned during the pretraining phase and these models struggle to on calculation-intensive tasks. Further one of the more important contributions of this work includes the massive multi-task dataset that comprises 57 different subjects.

1. This study shows how far off these language models are when compared to how humans use knowledge and commonsense reasoning to solve tasks. Along with that the collecting a comprehensive multi-task dataset that covers the depth and breadth of the multiple subjects would help the community understand these models better.
2.  One important comparison that would have definitely helped is trying to understand the gaps in knowledge between a GPT-3 to Unified QA. The difference in performance across the tasks seems high especially comparing an 11B to 175B.

Questions:
1. How would someone reproduce these experiments considering that widespread access GPT-3 models are not available to the general research community?
2. How was the level of difficulty of a subject measured? Was it already available when the datasets were being obtained?
3. Can you elaborate on the results from table 1 which shows that an 11B UnifiedQA model outperforms 175B GPT-3 model? Are these statistical significant?


===================== After reading the rebuttal ========
I thank the authors for providing further information and answering the question raised by the reviewers. Based on the responses and clarifying the issues I had, I have adjusted my score accordingly and improved it from a 5 to 6.

---

> ### Author Response · Authors · 2020-11-16
> **Reproducibility and Statistical Significance**
>
> Reviewer 1, thank you for your careful analysis and insightful suggestions. We hope the following clarifications address your concerns.
>
> **UnifiedQA vs GPT-3.**
>
> UnifiedQA is trained on various multiple-choice formats, whereas GPT-3 is not explicitly trained to handle multiple-choice questions. We believe this may explain the discrepancy in accuracy that we observe, and this effect may manifest in other multiple-choice datasets as well, which would be interesting for future work to explore. In the revised paper we have added GPT-2 results. The fine-tuned GPT-2 model also does worse than UnifiedQA, which suggests UnifiedQA/T5’s pretraining dataset C4 might be a cause of high performance. https://openreview.net/pdf?id=d7KBjmI3GmQ#page=5
>
> **Reproducibility.**
>
> We will release code for reproducing our experiments with the final version of the paper. Although the GPT-3 API is not widely accessible, UnifiedQA models and other models that we evaluate in the Appendix, such as RoBERTa-base and ALBERT-xxlarge, are available. In addition, we will release example-wise results for GPT-3 to help with reproducibility.
>
> **Difficulty levels.**
>
> We assign a difficulty level to a subject based on the progression of students in education systems. For instance, high school subjects are less difficult than college subjects, which are less difficult than professional subjects. This is justified, because professional subjects often require background from college education. Likewise, college subjects often require background from high school education.
>
> **We now have estimates of human-level performance.**
>
> Following your suggestion and that of other reviewers, we have evaluated human non-specialists on our benchmark using Amazon Mechanical Turk. Human non-specialist accuracy on the benchmark is 34.5%. We also estimate expert-level human performance as 89.8% by using the 95% percentile performance on subjects exams where statistics are available and by making educated guesses in other cases. We have added these results to the paper.
>
> **Statistical Significance.**
>
> Using Hoeffding's inequality, we note that if $n = 14079$, then with probability $1-0.05$, the absolute deviation to the true 0/1-loss is bounded by $\varepsilon$, where $\varepsilon$ is constrained by $n = \log(2/0.05)/(2\varepsilon^2)$. Note $\varepsilon=1.1$%, so the increase from 43.9% (GPT-3 few-shot) to 48.9% (UnifiedQA) is outside the 95 percent confidence interval. Hence the improvement is statistically significant, even highly statistically significant.
>
> We hope we were able to address your valid questions and we thank you for your helpful suggestions. Do you have any remaining concerns?

---

> > ### Comment · AnonReviewer1 · 2020-11-22
> > **Update**
> >
> > Authors,
> > Thanks for answering my question and running more experiments to make the results of this work reproducible. After reading the responses, I have adjusted my scores accordingly

---

### Official Review · AnonReviewer3 · 2020-10-27
**Recommendation to Reject**

**Rating:** 5
**Confidence:** 4

**Review:**

Summary:

The paper proposes a benchmark for NLP models. The purpose of this test is to measure the model's knowledge in 57 topics covered by approx 15000 tasks in total, each formulated as a closed-form question in zero-shot and few-shot settings. Most of the tasks were taken from different human examination sets. Then, the paper provides results of experiments with the latest (GPT-3 and T5 based) models along with some quantitative and qualitative observations.


Pros:
+ Such a dataset of questions may be useful in the future if it will be published.
+ The authors show that in the case of GPT-3, as with earlier similar models, the model's confidence is not a good estimate of the actual probability of the prediction is correct.

Cons and questions:
- The paper compares only models of two different architectures, while the proposed format for comparing the probabilities of 4 tokens allows on to test even models of the BERT family.
- I believe it's important to understand the human level in such benchmarks. Moreover, it would be interesting to see several human baselines with different levels of education.
- It is known that the ability to answer zero-shot/few-shot questions depends on the size of the model, but the inability to answer in the zero-shot format does not necessarily mean a lack of necessary knowledge in the model. Thus, the proposed approach is biased towards larger models that are just better able to work in zero-shot mode.
- Despite the fact that the UnifiedQA model is superior in quality to GPT-3, most of the results are devoted to the GPT-3 model.
- The uneven success of models for different topics can possible be explained by several objective reasons (the average length of a topic question in tokens, the average frequency of topic question tokens in the corpus, the share of topic documents in the training sample, and so on). At least some of them can be checked within the paper.
- It's unclear if the authors are going to publish the questionnaire.

Overall, I vote for rejecting. I like the idea of a comprehensive questionnaire as an NLP benchmark but besides the proposed set of questions the paper includes only the results for two different architectures and some debatable hypotheses on the reasons for such results.

UPDATE:
Authors, thanks for your updates.
Some of cons are gone, and the paper is better now, but my main concern stays: it's unclear if the results are about the problem solving ability or the zero-shot learning ability. Thus, I corrected my score to from 4 to 5.

---

> ### Author Response · Authors · 2020-11-16
> **Added GPT-2 Results and Question Length Analysis**
>
> Reviewer 3, thank you for your careful analysis and insightful suggestions. We hope the following clarifications address your concerns.
>
> **Comparisons to other architectures are in the Appendix.**
>
> In the main paper, we focus on T5/UnifiedQA and GPT-3 because they are state-of-the-art. However, we do in fact evaluate models of the BERT family (RoBERTa-base and ALBERT-xxlarge) in Section A.1 of the originally submitted paper. We have now also added results with GPT-2 in the revised paper per your suggestion. https://openreview.net/pdf?id=d7KBjmI3GmQ#page=5
>
> **We now have estimates of human-level performance.**
>
> Following your suggestion and that of other reviewers, we have evaluated human non-specialists on our benchmark using Amazon Mechanical Turk. Human non-specialist accuracy on the benchmark is 34.5%. We also estimate expert-level human performance as 89.8% by using the 95% percentile performance on subjects exams where statistics are available and by making educated guesses in other cases. We have added these results to the paper.
>
> **Fine-tuning results are in the Appendix.**
>
> Please see the Appendix for experiments with smaller, fine-tuned language models (RoBERTa, ALBERT). As mentioned above, we have added larger models to this comparison, such as GPT-2.
>
> **UnifiedQA vs GPT-3.**
>
> We have integrated the UnifiedQA results and GPT-3 results in the main paper by merging figures 6 and 9. We see the UnifiedQA results as demonstrating the benefit of fine-tuning on the MCQ format specifically, which may explain the improvement over GPT-3. The fine-tuned GPT-2 model also does worse than UnifiedQA, which suggests UnifiedQA/T5’s pretraining dataset C4 might be a cause of high performance. We analyze GPT-3 for declarative knowledge since it is few-shot, and UnifiedQA outputs a block of text instead of a single answer prediction token, so calibration assessment is not straightforward, unlike with GPT-3. We show UnifiedQA results adjacent to GPT-3 in Figure 6 in the revised paper. https://openreview.net/pdf?id=d7KBjmI3GmQ#page=6
>
> **Other explanatory factors.**
>
> We have added experiments exploring the effect of the average length of a topic question. For questions longer than a tweet (280 characters), the correlation between question length and true label confidence is slightly positive. This shows that longer questions are not necessarily harder. https://openreview.net/pdf?id=d7KBjmI3GmQ#page=13
>
> **Availability of the benchmark.**
>
> Should this paper pass peer review, we will publish the questionnaire and evaluation code.
>
> We hope we were able to address your valid questions and we thank you for your helpful suggestions. Do you have any remaining concerns?

---

### Official Review · AnonReviewer2 · 2020-10-28
**Interesting work on multitask performance of language models**

**Rating:** 8
**Confidence:** 4

**Review:**

In this paper, the authors propose a new test to measure a text model’s multitask accuracy based on 57 different tasks in Humanities, Social Science and STEM subjects. The authors experiment with the GPT-3 models of various sizes and UnifiedQA, and the results suggest that the size of the model may be one of the important factors in achieving higher performance on all the tasks.

This is an interesting work that tries to tackle a very important problem of how successful are large language models across multiple tasks. It would help the paper to have a more thorough discussion of the results. Based on the reported results, larger GPT-3 models perform better but a smaller UnifiedQA model outperforms GPT-2 by a substantial margin, why is this?

In addition, it would be interesting to see a comparison of GPT to BERT and XLNet. Such a comparison would first of all show if the idea generalizes to other types of language models. Second, it would emphasize the advantages/disadvantages of autoencoder-based vs autoregressive models and could potentially provide additional insights on how important these differences are for different tasks.

Given the authors provide additional discussion, I would like to see this paper and the associated dataset to be presented at the conference.

---

> ### Author Response · Authors · 2020-11-16
> **Fine-tuned GPT-2 Results Added**
>
> Reviewer 2, thank you for your careful analysis and insightful suggestions. We hope you will champion our paper. We also hope the following clarifications address your questions.
>
> **UnifiedQA vs GPT-3.**
>
> UnifiedQA is trained on various multiple-choice formats, whereas GPT-3 is not explicitly trained to handle multiple-choice questions. We believe this may explain the discrepancy in accuracy that we observe, and this effect may manifest in other multiple-choice datasets as well, which would be interesting for future work to explore. While we cannot fine-tune GPT-3 on multiple choice questions, we fine-tuned GPT-2 and added results to the main body (in addition to the RoBERTa and ALBERT results from the original submission, which were previously in the appendix). The fine-tuned GPT-2 model also does worse than UnifiedQA, which suggests UnifiedQA/T5’s pretraining dataset C4 might be a cause of high performance. https://openreview.net/pdf?id=d7KBjmI3GmQ#page=5

---

### Official Review · AnonReviewer4 · 2020-10-29
**Unsure if we should be testing for expert-level performance on 57 complex tasks.**

**Rating:** 6
**Confidence:** 3

**Review:**

This paper describes a dataset consisting of ~14k multiple-choice questions drawn from many different fields across the humanities and science as well as professional disciplines such as law and medicine. It presents results for GPT-3 models (LMs trained on text corpora with document context) of different scales, as well as for the UnifiedQA model (seq2seq model trained on various QA datasets). Performance of these models is well below their performance on other benchmarks: not above chance for the smaller GPT-3 models, and under 50% average accuracy for the best models.

The paper has a lot of good features: it’s obviously great to have a broad, challenging, large-scale dataset that aims to separate human from model performance. The lack of fine-tuning is a plus, as is having an objective evaluation procedure via multiple-choice answers. The calibration results for GPT-3 are very interesting.

However, I wonder about the value of a test that covers so many specialized areas of knowledge. A model that achieved expert performance (90% is suggested) across the board would clearly be superhuman, since no one is an expert in all these areas. Even a model that achieved expert performance in a few areas would probably be doing better than the average person. So instead of actually separating model performance from human performance, this benchmark might amount to just moving the goalposts about what constitutes human performance. To calibrate, it would have been helpful (albeit expensive, I realize) to see human scores from both experts and non-experts on this data.

Another problem is that the benchmark is likely to reward exposure to specialized domain data. Some tasks may benefit from this more than others, depending on the extent to which they are already represented in general-domain training corpora, and the extent to which they reward rote learning as opposed to true generalization. So it’s possible that the benchmark can be gamed by discovering these ‘easy’ tasks and doing aggressive data mining. The authors provide some evidence that this kind of approach isn’t likely to be fruitful for law, but that’s only one domain, and their experiment isn’t strictly comparable with, for instance, retraining GPT-3 on a corpus that includes many more legal documents. Ideally the paper would have included more evidence about the relative difficulty of tasks, linked to the amount of relevant training data for the models tested. Another option would have been to include only ‘hard’ tasks. But that begs the question about the value of using specialized tasks in the first place, as opposed to just training on general text and (more)  assiduously testing for world knowledge that’s unlikely to be explicitly represented (to borrow an example from the T5 paper: can a tuba fit into a backpack?).

Detailed comments and questions:
- Figure 2 runs up against the previous text.
- It would be good to merge UnifiedQA results (from figure 9) into figure 6.
- Is the plan to release this dataset publicly? There is no mention of this in the paper.
- To ensure a fair comparison, it would be interesting to run GPT-3 in the same text-generation mode as UnifiedQA.

---

> ### Author Response · Authors · 2020-11-16
> **We Test Specialization Since Knowledge Is Valuable**
>
> Reviewer 4, thank you for your careful analysis and insightful suggestions. We hope the following clarifications address your concerns.
>
> **Reasons for testing on numerous, complex tasks.**
>
> We believe testing models on complex tasks is useful, because economically valuable applications are often complex and require high school knowledge or more. Measuring how well models learn these tasks from pretraining material is important for creating an extensible general purpose technology for numerous downstream applications. Likewise, this task can help pinpoint areas in which models, datasets, and training schemes are lacking, paving the way for ‘well-rounded’ pretrained representations or broadly applicable mixture of experts models.
>
> Note that our benchmark is not wholly composed of complex tasks, as we also include elementary and high-school level subjects.
>
> **We now have estimates of human-level performance.**
>
> Following your suggestion and that of other reviewers, we have evaluated human non-specialists on our benchmark using Amazon Mechanical Turk. Human non-specialist accuracy on the benchmark is 34.5%. We also estimate expert-level human performance as 89.8% by using the 95% percentile performance on subjects exams where statistics are available and by making educated guesses in other cases. We have added these results to the paper.
>
> **Rote learning vs generalization / gaming the benchmark.**
>
> We agree that some of the 57 subjects are more amenable to memorizing facts from textbooks. However, many of the subjects, including computer science, mathematics, and professional law, require applying knowledge to new, combinatorially diverse scenarios. Even subjects that are memorization-heavy tend to phrase exam questions in a different way from the available learning material. This requires models to demonstrate understanding.
> To the extent that aggressive data mining can help round out the datasets these models are trained on, we consider it a valid approach. However, our professional law experiments in Section 5 suggest that this approach may not scale for specialized subjects, because there may not be enough data. We see this as a reason to focus on data efficiency in contrast to the prevailing doctrine of collecting a bigger training set.
>
> Since we test such a broad array of tasks, our test is unlike NLP benchmarks that test a narrow range of concepts; it is far less likely to be wholly solved through simple gaming tactics. Additionally, in Figure 12 we show that log-probability of the questions on each task does not correlate with accuracy on the tasks, suggesting that memorization is not currently an issue.
>
> **Comparison to commonsense benchmarks.**
>
> Physical commonsense questions such as “can a tuba fit into a backpack?” demonstrate knowledge of object sizes and relationships, but the questions in our benchmark require a deeper understanding of societally/vocationally important pieces of knowledge that educators have decided to put in their courses and textbooks. Models that read the Internet are likely to be exposed to this vast range of knowledge, so evaluating their mastery on a broad range of subjects is a natural next step.
>
> **Details.**
>
> We do plan to release the dataset and experiment code publicly.
> Thank you for your suggestion to merge Figures 9 and 6. We have done so in the updated paper. https://openreview.net/pdf?id=d7KBjmI3GmQ#page=6
>
> We hope we were able to address your valid questions and we thank you for your helpful suggestions. Do you have any remaining concerns?

---

> > ### Comment · AnonReviewer4 · 2020-11-24
> > **Still unsure**
> >
> > Thanks for the careful and informative response, and for the improvements you made to the paper. I’m afraid I’m still really not convinced that this test is measuring something meaningful connected to human understanding or knowledge acquisition, especially since the Mechanical Turk data seems to show that the LMs are above average human performance. But on the other hand, I’m not convinced that it will be useless, so I’m giving you the benefit of the doubt and changing my score to leaning positive.

---

### Author Response · Authors · 2020-11-18
**New Version of the Paper**

A revised paper has been uploaded which aims to address several reviewer comments. In the update, we

- added a Amazon Mechanical Turk human-level baseline
- merged GPT-3 results with the UnifiedQA results into a single figure https://openreview.net/pdf?id=d7KBjmI3GmQ#page=6&zoom=100,408,300
- added analysis of question length vs difficulty in the appendix
- added RoBERTa and ALBERT results to the main body in Table 1 (these results were in the appendix of our original submission)
- added GPT-2 results https://openreview.net/pdf?id=d7KBjmI3GmQ#page=5&zoom=100,144,768

---

### Comment · ~Dan_Hendrycks1 · 2021-01-20
**Dataset and Code Now Available**

The dataset and code is available here: https://github.com/hendrycks/test

---

### Decision · Program_Chairs · 2021-01-07
**Final Decision**

**Decision:**

Accept (Poster)

**Comment:**

This paper presents an extensive evaluation of two language models: GPT-3 and UnifiedQA on 57 tasks. The results demonstrate that these models are still far from expert-level accuracy and do not know when they are wrong.

I think this is an interesting paper that provides useful insights into the capability of large-scale language models. The authors also plan to release their dataset and have addressed some of the concerns from the reviewers to improve the paper during the rebuttal period.